# Unsupervised Discovery of Dynamic Neural Circuits

Colin Graber      Ryan Loh      Yurii Vlasov      Alexander Schwing
University of Illinois at Urbana-Champaign
{cgraber2, ryanloh2, yvlasov, aschwing}@illinois.edu

## Abstract

What can we learn about the functional organization of cortical microcircuits from large-scale recordings of neural activity? To obtain an explicit and interpretable model of time-dependent functional connections between neurons and to establish the dynamics of the cortical information flow, we develop '*dynamic* neural relational inference' (dNRI). We study both synthetic and real-world neural spiking data and demonstrate that the developed method is able to uncover the dynamic relations between neurons more reliably than existing baselines.

## 1 Introduction

Extraction of latent temporal dynamics in complex networks is important to understand their functional connectivity and to predict their behavior. Recently, various machine learning methods were used to encode/decode the behavior from recorded activity of large neuronal populations [2, 3]. However, in these mostly 'static' brain models the temporal dynamics of the firing activity as well as interactions between different neurons are often neglected. It is expected, however, that the dynamic interactions in neural networks might be the key to understanding the brain computations. Addressing this, several methods have been proposed to uncover low-dimensional latent representations of neural network activity and its dynamics, including dimensionality reduction-based techniques such as principal components analysis [1] and tensor components analysis [14], pattern extraction techniques based on matrix factorization such as ConvNMF [11] and SeqNMF [7], and autoencoder models such as LFADS [9]. However, temporal correlations between individual neurons in the network are often only modeled implicitly, hindering reconstruction of functional connectivity of the neural circuits.

In contrast to these implicit techniques, here, we develop an extension to Neural Relational Inference [6], which we call 'dynamic Neural Relational Inference' (dNRI). Specifically, we develop a new model to extract rapid dynamic changes of network activity in the form of a time-dependent adjacency matrix. We aim at extracting rapid (tens of milliseconds) correlations between recorded neurons that capture their functional relations across the network.

Moreover, our method enables the tracking of the temporal evolution of this functional connectivity over the span of a trial. This means it can provide an interpretable approach to uncover hidden dynamical structure of brain information flows and to reconstruct the underlying functional brain circuitry. We demonstrate the applicability of our method on both synthetic spiking data and data recorded from the cortex of live and behaving mice.

## 2 Dynamic Neural Relational Inference (dNRI)

We are interested in recovering the dynamic flow of information between neurons, *i.e.*, we want to estimate whether spiking of one neuron either excites or suppresses spiking of another neuron at various points in time. To address this task, we assume spiking information for a set of neurons to be available. We represent neural spiking information via matrices $\mathbf{x} \in \{0, 1\}^{N \times T}$, where $N$ is the number of neurons recorded for $T$ time bins and each entry represents the absence or presence of a spike for a particular neuron $i$ at a given time bin $t$. The goal is to predict binary variables $\mathbf{z}_{ij}^{(t)}$ (hereafter called 'edges') for every pair $(i, j)$ of neurons for every timestep $t$ which indicate whether the spiking activity of neuron $i$ influences that of neuron $j$. With the assumption that neurons $i$ and $j$ are connected, setting $\mathbf{z}_{ij}^{(t)} = 1$ indicates that this connection is currently 'active' at time $t$.

33rd Conference on Neural Information Processing Systems (NeurIPS 2019), Vancouver, Canada.

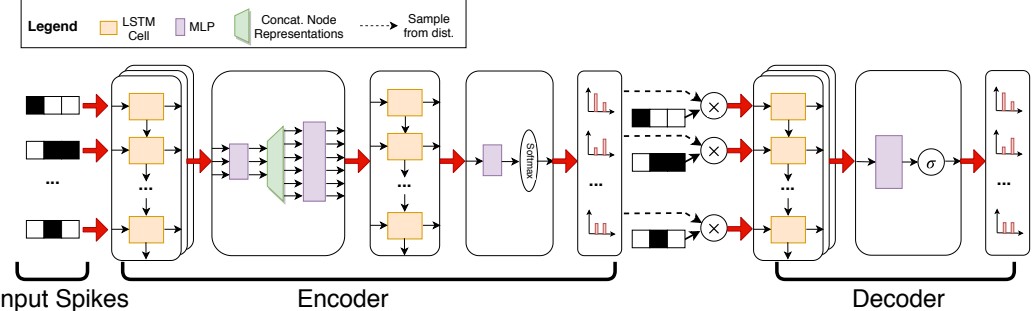

Figure 1: Our dNRI model: an encoder which processes neural spiking data to produce edge probabilities for every time step and a decoder which uses samples from this distribution to reconstruct the original spiking data.

To model this problem, we follow the recently introduced NRI formulation [6] and learn a variational auto-encoder (VAE) whose observed variables represent neural spiking patterns $\mathbf{x}$ and whose latent variables $\mathbf{z}$ represent connections between neurons. See Fig. 1 for a depiction of the full model. Unlike Kipf et al. [6], who focus on predicting static connectivity graphs, we model dynamic connections that vary across time. Additionally, they optimize the evidence lower bound (ELBO), but we instead use the $\beta$-VAE formulation described by Higgins et al. [4]. More formally, we optimize the following variational objective

$$\mathcal{L}(\phi, \theta) = \mathbb{E}_{q_\phi(\mathbf{z}|\mathbf{x})} \left[ \log p_\theta(\mathbf{x}|\mathbf{z}) \right] - \beta \mathrm{KL}[q_\phi(\mathbf{z}|\mathbf{x})||p(\mathbf{z})]. \tag{1}$$

This objective consists of three major components, which we will describe subsequently.

**Encoder.** The encoder $q_\phi$ takes an entire neural spike train $\mathbf{x}$ as input and produces $q_\phi(\mathbf{z}|\mathbf{x})$, which is an approximate posterior probability distribution for each connection variable. The encoder hence estimates the probability of a neuron $i$ being connected to neuron $j$ at time $t$. We use long-short-term-memory (LSTM) unit-based deep nets parameterized by $\phi$ as our encoder. $q_\phi$ is then used to sample likely interaction patterns which are used by the decoder. Because the latent variables $\mathbf{z}$ are discrete, the process of sampling from their distribution is non-differentiable. Consequently, we follow prior work and sample instead from the concrete distribution [8, 5], which approximates discrete sampling in a differentiable manner and enables to backpropagate gradients from the decoder reconstruction all the way to the encoder parameters $\phi$.

**Decoder.** The decoder $p_\theta(\mathbf{x}|\mathbf{z})$ models the probability of reconstructing the input spike train given a sampled set of edges from the approximate posterior. For this we also use an LSTM unit-based deep net and refer to its parameters via $\theta$. A separate recurrent neural net (RNN) is used to model each neuron. To represent the influence of the predicted edges, these RNNs take as input a *masked* version of the predicted spiking probability for every neuron from the current time step, where the mask for each neuron is derived from the sampled edges.

**Prior.** The choice of the prior $p(\mathbf{z})$ is used to encourage sparsity of the modeled edges. Because we want edge predictions to be independent of each other, we use an independent Bernoulli prior $p(\mathbf{z}) = \prod_{t=1}^{T} \prod_{i \neq j} p(z_{i,j}^{(t)})$ for each latent variable. Setting the probability of no edge (*i.e.*, $p_\theta(z_{i,j}^{(t)} = 0)$) larger than 0.5 reduces prediction of spurious edges. On synthetic data, we found that using a value of 0.8 worked well for our experiments. For the real data, however, we found that using a strong no-edge probability prevented the model from picking up the relatively sparse connections, so we used a uniform prior for the experiments on real-world spiking data reported below.

**To train the parameters** $\theta$ and $\phi$ of the decoder and encoder, we proceed as follows: for each spike train in the current minibatch, the encoder first predicts the approximate posterior $q_\phi(\mathbf{z}|\mathbf{x})$ for each latent variable. We then sample from this distribution as discussed previously. Given these samples $\hat{\mathbf{z}}$, we then predict spiking activity using the decoder $p_\theta(\mathbf{x}|\hat{\mathbf{z}})$. For training, we use ground-truth spikes as the decoder input; during testing, predictions for each time step are fed as input into the next step.

## 3 Experiments

We demonstrate the efficacy of dNRI using two types of data: the first are three synthetic datasets consisting of 12 simulated neurons with baseline spiking rates each sampled from the interval $[0.1, 0.3]$. Additional spikes are generated as follows: time is divided into four phases, with each phase containing 10 randomly sampled neuron pairs $(i, j)$ which indicate that whenever neuron $i$

GT    dNRI    GLM    TCA    seqNMF

Phase 1 / Phase 2 / Phase 3 / Phase 4

Figure 2: Average predicted edges for each phase for test data with 0.8 edge probability. From left to right, the plots are for the ground-truth, dNRI, GLM, TCA, and seqNMF.

Table 1: Metrics computed on synthetic data

| Edge Prob. | Method | Edge F1 | | | Reconstr. Error | | |
|---|---|---|---|---|---|---|---|
| | | Train | Val | Test | Train | Val | Test |
| 1 | TCA | 32.2 | 33.2 | 32.5 | 0.431 | 0.402 | 0.416 |
| | seqNMF | 23.8 | 26.8 | 23.7 | 0.013 | 0.019 | 0.013 |
| | GLM | 43.2 | 43.6 | 42.7 | 0.749 | 0.722 | 0.736 |
| | Static (d)NRI | 43.7 | 43.7 | 43.7 | 0.988 | 0.948 | 0.989 |
| | Ours (dNRI) | **82.5** | **81.5** | **80.0** | 0.977 | 0.930 | 0.970 |
| 0.8 | TCA | 30.8 | 33.1 | 33.7 | 0.479 | 0.449 | 0.465 |
| | seqNMF | 21.1 | 25.3 | 22.2 | 0.010 | 0.011 | 0.010 |
| | GLM | 44.0 | 44.3 | 41.3 | 0.790 | 0.764 | 0.778 |
| | Static (d)NRI | 43.7 | 43.7 | 43.7 | 1.052 | 0.982 | 1.034 |
| | Ours (dNRI) | **89.1** | **88.2** | **84.4** | 0.970 | 0.948 | 0.997 |
| 0.6 | TCA | 29.7 | 32.8 | 32.7 | 0.522 | 0.493 | 0.510 |
| | seqNMF | 17.4 | 22.2 | 20.1 | 0.008 | 0.008 | 0.008 |
| | GLM | 44.4 | 44.3 | 42.4 | 0.824 | 0.799 | 0.813 |
| | Static (d)NRI | 38.0 | 38.0 | 38.0 | 0.967 | 0.926 | 0.972 |
| | Ours (dNRI) | **89.7** | **87.6** | **84.9** | 0.918 | 0.901 | 0.931 |

spikes at time $t$, neuron $j$ will spike at time $t + 1$ with probability 1, 0.8 or 0.6 (hereafter referred to as 'edge probability'). The second type of data consists of spiking activity of 24 neurons which are binned at 20 ms and recorded from a primary somatosensory cortex of a mouse actively navigating in a tactile virtual reality while motorized walls were moved towards and away from the animal snout [13, 12]. We compare the proposed approach on the synthetic data to four baselines using the following metrics: their ability to find the underlying edges (measured via F1) and on the normalized reconstruction error of neural spiking, computed as $\|\hat{x} - x\|_F / \|x\|_F$ where $x$ is the original spiking data, $\hat{x}$ is the predicted reconstruction and $\| \cdot \|_F$ denotes the Frobenius-norm. Each dataset is separated into train, validation, and test splits, with dNRI models being trained on the train split and hyperparameters being tuned using performance on the validation split. Test set results are presented.

We use the following baselines: **Tensor Component Analysis (TCA)** [14] is a PCA extension that factorizes the data into time, trial, and neuron components. We first convolve input data with a Gaussian filter. After running TCA, we take the outer product of neuron factors with themselves to find neurons that spike close to each other. Predicted edges are then obtained by multiplying this result by the time and trial components to get predictions at each time step. **SeqNMF** [7] is an extension of non-negative matrix factorization that produces a matrix factor representing neural activity for some fixed length of time and a vector factor representing time. To predict edges from learned factors, we take the outer products of all columns of the neuron factor, which produces edge matrices whose values are large for neurons that spike in sequence. We multiply these by their corresponding time factors to get predictions per time step, and sum the contributions from each factor to obtain final edges. **Static (d)NRI** employs a static graph dNRI model where the outputs of the encoder edge LSTM are averaged across time before computing the final edge probabilities. These probabilities are then used for all time bins. **GLM** consists of a Bernoulli generalized linear model [10] per neuron, using all neuron spiking history as covariates. Note that not all baselines were originally developed for this task, yet we think they are applicable.

**Synthetic Data Results.** The computed metrics for all of the synthetic datasets for all models are reported in Tab. 1. None of the baselines are able to recover the dynamic connections reliably. In contrast, dNRI is able to recover these interactions to a high degree of accuracy. Also note the benefits of dynamically estimating adjacency as opposed to a static interaction. Moreover, this performance is maintained when the edge spiking probability becomes smaller. Many of the baselines outperform dNRI at reconstructing the original spiking activity, but this is a consequence of the difference in training objectives or inference procedures. Fig. 2 visualizes the edge predictions made by dNRI.

**Mouse Cortical Recording Data Results.** In analysis of the real-world data, we focus on a choice period between 400ms from the start of the trial, when the animal starts to sense the approaching wall, and 950ms, when the animal is making a decision to change the run direction to avoid the approaching wall. We present the results on this data in Fig. 3 focusing on several frames that correspond to the last stage of sensory information processing when the animal has almost made a choice and is preparing for motor action. Neurons are ordered with respect to their cortical depth and

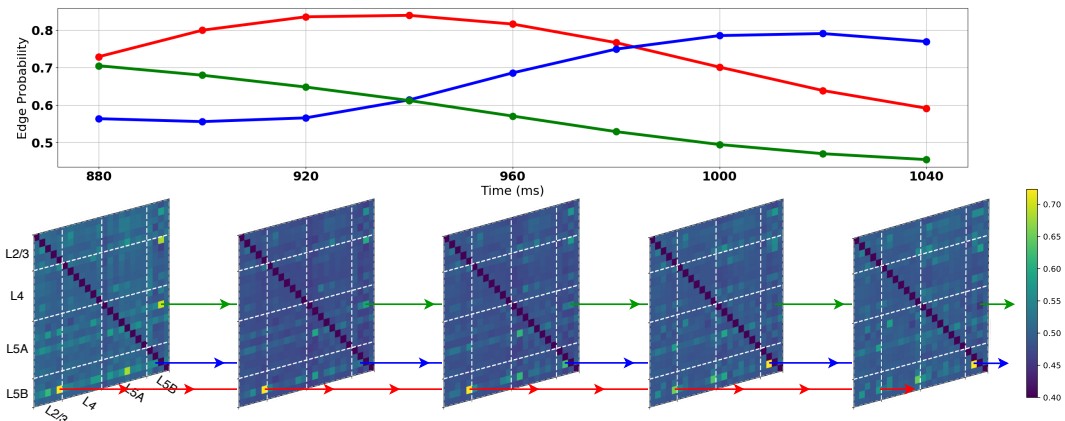

Figure 3: dNRI results for mouse cortex data recorded during animal choice period.

assigned to specific cortical layers. While the overall population spiking activity is relatively dense, significant correlations revealed by dNRI are sparse. This is expected, as we are focusing only on rapid correlations to reveal putative monosynaptic connections. Correlations are also transient, with a typical lifetime on the order of 90ms. In Fig. 3, we highlight several neuron pairs to exemplify the power of our representation: dNRI results infer transient information flow from L2/3 to L5B neurons (red curve) as well as communications within deep L5A and L5B (blue and green curves), as they are strongest outputs of the somatosensory barrel columns. Similar to the analysis of synthetic data trials, neither SeqNMF, GLM, nor TCA are able to capture fast transient features revealed by dNRI.

Fig. 4 displays the cross-correlations between the neuron pair whose predicted edges are highlighted in red in Fig. 3. The use of cross-correlations is a standard method of analysis used to discover putative monosynaptic connections between neurons. The results in Fig. 4 indicates the presence of such a connection; however, this sort of analysis does not provide any information regarding when this connection is being actively used in the network. As displayed in Fig. 3, the dNRI model was not only able to successfully detect the presence of this connection, but it also predicts when this connection is active. In other words, dNRI allows for additional analyses of neural spiking data that are not possible when using a static analysis technique.

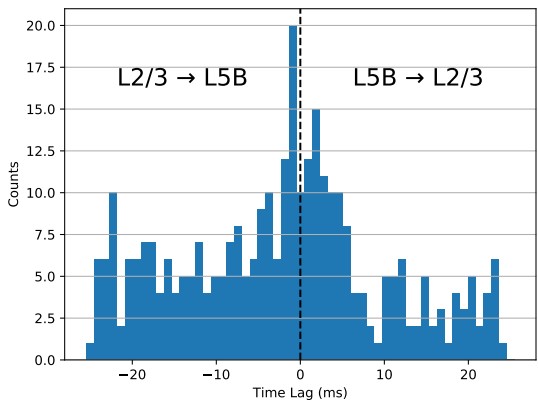

Figure 4: Static cross-correlation plot for neuron pair highlighted in red in Fig. 3.

## 4 Conclusions

We develop a method to explicitly extract *time-dependent* functional relations from large-scale neural spiking data recordings of cortical networks. Using simulated data of spiking activity where ground truth is available, and real data, we demonstrate that the proposed approach is able to recover the implanted interactions more accurately than baselines which model relations implicitly.

**Acknowledgments:** This work is supported in part by NSF under Grant No. 1718221 and MRI #1725729, UIUC, Samsung, 3M, Cisco Systems Inc. (Gift Award CG 1377144) and Adobe. We thank NVIDIA for providing GPUs used for this work and Cisco for access to the Arcetri cluster.

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
