# OpenReview forum: "Unsupervised Discovery of Dynamic Neural Circuits"
_NeurIPS.cc/2019/Workshop/Neuro_AI — Real Neurons & Hidden Units @ NeurIPS 2019 Poster_

### Official Review · AnonReviewer2 · 2019-09-23
**dNRI model presents an interesting new framework for investigating neural interactions**

**Clarity:** 5

**Comment:**

Strengths:
This is an interesting application of several AI techniques to an equally interesting and increasingly pressing problem in neuroscience. The model was well-motivated and well-described, and the synthetic data results are convincing. The authors did a reasonable job of comparing to other recent methods introduced in the neuroscience literature.

Areas for improvement:
I have two suggestions for improving the analysis, which will be helpful in convincing neuroscientists that this is a useful tool:
1) There are other models for determining dynamic functional connectivity that might be additional useful comparisons; for examples see Foti & Fox, "Statistical model-based approaches for functional connectivity analysis of neuroimaging data"
2) An interpretability issue arises when the network is only partially observed. Since this is almost exclusively the case in neuroscience, it would be interesting to see how this method performs when you simulate, say, 24 neurons and then only observe 12 of them. How does this change the inferred network structure?

**Category:**

AI->Neuro

**Clarity Comment:**

The model and results are well-described in the limited space provided.

**Evaluation:**

5: Excellent

**Importance:**

4: Very important

**Importance Comment:**

As recording technologies continue to increase the number of simultaneously recorded neurons across many model organisms, developing new techniques to understand how these neurons form local circuits (and hence what computations they perform) will be increasingly important.

**Intersection:**

4: High

**Intersection Comment:**

This model uses several recent advances in AI (VAEs, beta-VAEs, concrete distributions) to tackle a difficult neuroscience problem.

**Rigor Comment:**

The proposed model is an extension of several other well-known techniques, and incorporates them all cleanly.

**Technical Rigor:**

4: Very convincing

---

> ### Author Response · Authors · 2019-10-30
> **Thanks**
>
> Thanks for the kind comments and for sharing the insightful survey.

---

### Official Review · AnonReviewer3 · 2019-09-25
**A VAE-based approach for inferring dynamic functional connectivity between neurons**

**Clarity:** 5

**Category:**

AI->Neuro

**Clarity Comment:**

Every part of the study is described clearly. Despite the limited space provided, the authors have successfully managed to explain the concepts, theory, and results with sufficient detail and have not left any ambiguity in the text.

**Evaluation:**

4: Very good

**Importance:**

4: Very important

**Importance Comment:**

Latent dynamics of neural populations reflect the computations performed by the population. Therefore, inferring the these latent dynamics from noisy multiunit recordings is of great importance in system neuroscience. Building on top of recent advances in unsupervised deep learning, this paper proposes a novel method for inferring dynamic latent connectivity between single neurons based on recorded spiking activity of single units.

**Intersection:**

3: Medium

**Intersection Comment:**

The paper has employed several most recent concepts in unsupervised deep learning to tackle an important methodological problem in neuroscience.

**Rigor Comment:**

The theoretical concepts and the experimental results are sufficiently rigorous and convincing.

**Technical Rigor:**

4: Very convincing

---

> ### Author Response · Authors · 2019-10-30
> **Thanks**
>
> Thanks for the kind comments.

---

### Official Review · AnonReviewer1 · 2019-09-25
**VAE model performs impressively, but comparisons are specious**

**Clarity:** 3

**Comment:**

The author has proposed a very interesting model in this submission, but the same level of technical rigor deployed in model development has not been distributed to the many aspects of its testing and validation. I would not recommend this submission's acceptance in its current state.

**Category:**

AI->Neuro

**Clarity Comment:**

Putting a single number on this measurement is overly reductive in the case of this publication. The descriptions of the dNRI model, the optimization methods used to train it, and the data that the model was trained and tested on in the scope of this paper are clear and reasonably thorough. For that reason, I'd like to rate it highly; however, the descriptions of model test results that follow the laudably written sections lack the details required for reasonable comprehension. Many of these issues are more fully detailed in the "technical rigor" section, though the reviewer will also allude to them here, as the omission of these important details is a detriment to the paper's ability to cogently communicate the validity of this research project the reader. For that reason, I've split the difference at the with a weighted sum toward the lower of the two scores.

**Evaluation:**

2: Poor

**Importance:**

4: Very important

**Importance Comment:**

Finding unsupervised methods to accurately estimate network connectivity from output time series data is paramount to the analysis of complex systems with unknown network structure (e.g. neuronal data). The model presented in this paper is a good extension of the NRI methods it's built from and is shown to be relatively accurate in an F1 metric of estimated connectivity, but the poorly detailed comparison estimation methods and very low data reconstruction accuracy are problematic.

**Intersection:**

4: High

**Intersection Comment:**

The use of VAE models to estimate network structure is clearly very powerful, and the estimation of network structure from time series outputs of complex systems is an important and general open problem with many useful applications in the analysis of several modalities of neuroimaging data. This paper contains a detailed description of just such a model that is designed to function at small time scales capable of modeling such dynamic functional connectivity activity. While it does not make a good case for comparison to other supposedly state-of-the-art models, the potential for utility is quite high within the neuroscience data analysis community.

**Rigor Comment:**

The dNRI model proposed is well-detailed in concept: a VAE model whose encoder and decoder networks map from network node signal data to estimates of network connectivity (given as a probabilities) and vice versa, respectively. The network is able to produce network structure estimates that score highly on an F1 metric relative to other methods presented. This is impressive, but the results are not thoroughly presented or adequately qualified.

Some of the methods presented as viable comparisons of network estimation performance are questionable. The use of an NRI method is well-motivated, as the presented dNRI model is a direct development from a "static" or time-averaged NRI model. The use of GLM models from estimating spiking activity is not motivated beyond a passing citation that does not relate them to the other models presented. The use of Tensor Decompostion (low-rank canonical polyadic forms of trial-segmented multidimensional signal data) and the SeqNMF model to estimate network connectivity is interesting and partially relevant given this pair of models' ability to model dynamic changes in network activity patterns, but is not proper for technical reasons, the foremost and most objectionable of which being the authors' assumption that TDA's neuron activity dimension output and SeqNMF's canonical sequence firing output estimates can be used to estimate network connectivity at all.

In the former case, a inappropriately cursory description of the network connectivity estimation method is given as taking the dyadic product of the TDA neuron/channel vector and "convolving" the resultant connectivity matrix with the trial and time vectors. The second part of that statement, found in a single sentence of the second paragraph of section "3 Experiments," does not provide the clarity required to treat such a mathematical operation. The first part, regarding the use of the dyadic (outer) product of what are essentially signal component strength vectors to estimate network connectivity, is not appropriate. Showing that two "neurons" are coincidentally firing within a data segment window is not at all equivalent to the description of the network connectivity estimates produced from the dNRI model and is in its own right a questionable statement. A very top-level objection I have with this method is that it is only capable of producing square matrices, while figure 2 clearly shows that the ground truth networks tested in the model are directed graphs with non-symmetric association matrices. Furthermore, the order of the TDA method is not stated. If greater than 1, this would produce several individual network connectivity estimates. In the current publication, there is no mention of how these order vectors are combined.

The second of the two methods, the SeqNMF model, decomposes an input signal of superimposed firing sequences through deconvolution into a tensor of firing sequence atoms and a time signal of impulses representing their place in the recomposed signal. The authors have extended their interpretation of these outputs to consider that firing sequences imply an underlying network structure, and have apparently sought to estimate that structure by taking some form of an outer product of individual sequence matrices. The means by which they perform this lightly described method of network reconstruction is entirely unclear. This would suffer, the review must assume, from similar issues regarding the necessarily symmetric output of dyadic products over real-valued vector data; however, the apparent break in interpretation from what the reviewer understands as the correct understanding of SeqNMF model outputs is much more damaging to the publication's quality.

One further issue is that of the lacking treatment of input data reconstruction accuracy results. There is mention that "Many of the baselines outperform dNRI at reconstructing the original spiking activity, but this is a consequence of the difference in training objectives or inference procedures," but the exact nature of those differences should be explored much further. While the dNRI model is not the only model presented that is reporting upwards of 100% f-norm error metrics, it should be explored more clearly

**Technical Rigor:**

2: Marginally convincing

---

> ### Author Response · Authors · 2019-10-30
> **Response to comments and clarification of a few details**
>
> Thanks for sharing detailed thoughts.
>
> - Re. comparison: To the best of our knowledge no available method is directly applicable to the proposed task. Nonetheless we felt meaningful baselines could be obtained via minor modifications of existing methods, despite them not being originally designed and proposed for this task.  Widespread use in modeling (time-averaged) relations among spiking neurons (GLM) and the ability to model time-varying relations between neurons (TCA/seqNMF) guided our choice. The latter two methods do not directly model connections between neurons, but we did our best to extract connectivity.
>
> We would be very grateful for a hint if the reviewer is aware of better baselines to extract dynamics from brain activity.
>
> - Re. details: Apologies for brevity due to the page limit. For TCA, we convolved the data with a Gaussian filter across the time dimension (to model non-synchronous connections) before running TCA. Afterwards we took an outer product of each neuron factor with itself, which attempts to find neurons with neighboring spiking. We then multiply this new "edge" factor by time/trial factors to produce the edge predictions for each set of factors.
> For seqNMF, we use neuron factors containing a sequence length of two (since we know this is the temporal extent of ground-truth connections). Afterwards, for each set of factors, we take the outer product of the two columns of the neuron factor. This produces an "edge matrix" whose entries are large for neuron pairs that spike in sequence. We then multiply these edges by the time factor to get an edge prediction per time step.
> In both TCA and seqNMF, the final edge predictions are found by summing the contributions for each set of factors.
>
> - Re. reconstruction accuracy: we used 52 factors for both TCA and seqNMF, which represents the number of factors required  to perfectly represent each of the 40 ground-truth connections as well as  spiking of the 12 neurons. Therefore, it is possible to represent the data exactly (of course, the optimization procedure may not recover these parameters). The other methods (NRI, GLM) do not use separate parameters for each trial, which is why their reconstruction error is larger.

---

### Decision · Program_Chairs · 2019-10-02

Accept (Poster)